# Results of 15 Years of Precision Feeding of Hyper Prolific Gestating Sows

**DOI:** 10.3390/ani11102908

**Published:** 2021-10-08

**Authors:** Nathalie Quiniou

**Affiliations:** IFIP—Institut du Porc, BP35104, CEDEX, 35651 Le Rheu, France; nathalie.quiniou@ifip.asso.fr; Tel.: +33-2-99-60-98-38

**Keywords:** sow, gestation, precision feeding, energy, monitoring, backfat thickness

## Abstract

**Simple Summary:**

Body condition at farrowing influences the short-term performance of sows (e.g., farrowing progression, milk production, appetite during lactation) and their long-term reproduction and longevity potentials. In the context of increased prolificacy, farmers increasingly focus on managing maternal reserves. Measurements of backfat thickness and body weight at insemination can be used to monitor the feed allowance during gestation and to maintain body condition throughout farrowing. Regular analysis of data collected on farms can identify the ideal backfat thickness and expected body weight at farrowing (as a function of age). Model-based assessment of nutritional requirements can consider housing conditions, prolificacy, initial characteristics of the sow at insemination (i.e., age, body weight and backfat thickness) and final expected characteristics. On a demonstration farm, feed was provided on an individual basis based on the energy requirements calculated for each sow during gestation under different housing conditions. The risk of high variability in backfat thickness within each batch was reduced at farrowing.

**Abstract:**

The increase in prolificacy at weaning is less than that at farrowing due to increased loss of piglets. As a result, farmers focus more on solutions that can prevent difficult parturition or a decrease in milk production. The body condition of the sow influences both factors. A model developed to estimate energy requirements of gestating sows was used to monitor the body weight and back fat thickness (BT) at farrowing, through the creation of a demonstration farm that included 7 batches of 24 sows. Daily feed allowance was adapted to characteristics of each sow at the beginning of gestation. Based on data collected since 2005 from 5140 gestations in different housing systems, the BT averaged 19.3 mm at farrowing, with no significant differences among housing systems. Within-batch variability in BT ranged from 3–4 mm and is expected to improve in the future by using sensors to automatically weigh and measure physical activity towards a real-time assessment of energy requirements. The next step in reducing feed costs and environmental impacts is to consider amino-acid and phosphorus requirements in the precision-feeding strategy.

## 1. Introduction

Increasing the prolificacy of sows in a breeding-fattening unit increases farmers’ incomes through the number of fattened pigs sold per reproductive animal. Consequently, genetic-selection companies have continuously focused their efforts on this criterion for decades. In France, 1994 is considered the year in which hyperprolific sows were distributed to nearly all commercial farms. Soon after, farmers began to complain about the lack of progress in performance from birth to weaning. Increased loss rates were reported [1] that may have been related to certain changes in characteristics of the litters. The increase in total birthweight of the litter did not appear to be proportional to the increase in litter size, which resulted in lighter piglets at birth, on average. An increase in within-litter variability in birthweight was also reported, with a low survival rate of the lightest piglets before weaning [2,3]. Given this context, more emphasis was placed on parturition and the subsequent milk production. As both criteria are closely connected to the body condition of the sow at farrowing [4,5], it was recommended to revise the feeding strategy on pig farms during gestation or during the entire reproductive cycle. This revision included the total amount of feed supplied during gestation as well as long-standing feeding plans, as the increase in litter birthweight from hyperprolific sows was associated with additional nutritional requirements during gestation [6]. This was especially apparent during late gestation when foetuses gain the most weight [7]. 

Improving the balance between nutrient supplies and nutrient requirements during gestation was expected to improve the management of sows within herds. When the first reproductive sows arrived at the demonstration farm used in this study (Romillé, France, built in 1997), feed delivery was adapted to characteristics of each sow at the beginning of gestation: age (Age_initial_, day), body weight (BW_initial_, kg) and backfat thickness (BT_initial_, mm). To this end, the modelling approach developed at INRAE over previous decades was used, based on published equations available at that time. This article shows how this precision-feeding strategy was implemented and used to monitor the backfat thickness at farrowing (BT_final_) using individualized feed supplies during gestation for nearly 25 years.

## 2. Materials and Methods

### 2.1. Description of the Demonstration Farm

The demonstration farm manages seven batches (labelled “1” to “7”) of 26–30 crossbred Large White × Landrace sows at mating to keep 24 at farrowing, with a 3-week interval between matings and piglets weaned at four weeks of age. After weaning, the label given to the batch is changed to represent the number of times the batch was reproduced (the “reproductive cycle” of the farm). For example, the number 536 is given to batch 6 of the 53rd reproductive cycle. From 1997–2013, sows were either kept in stalls after insemination (three batches) or moved to pens containing six sows (two batches) or 12 sows (two batches). As of 1 January 2013, European Union regulations (Article 3(4) of Directive 2008/120/EC on the protection of pigs) no longer allowed sows to be kept in stalls throughout the entire gestation period; subsequently, reproductive sows were kept in stalls for less than 4 weeks after service. The sows were scanned 25 days after the first artificial insemination. Pregnant sows were moved to pens containing four to six sows (two batches) or to a large pen designed to house 72 sows, coming from three batches present simultaneously, and used to house five batches.

All sows were fed individually in these housing systems. The individual daily feed allowance was weighed manually and delivered in two meals per day (except on Sundays, which had only one meal) in an individual trough for sows housed in stalls or in small groups. During the meal, the sows housed in small groups were locked in mobile baskets, were identified by their ear tag and then received their individual allowances. In contrast, pens of 12 or 72 sows were equipped with one or two electronic feed dispensers, respectively. When a sow entered the feeding station, an antenna detected and identified its RFID ear tag, and its feed allowance, as stored in the computer, was delivered. Sows from seven batches were not fed this way when the gestating rooms were under reconstruction in 2013. The solution was to keep the gestating sows in rooms in the fattening unit, where it was not possible to control the individual feed allowance.

During gestation, sows were fed a gestation diet in the form of pellets. The nutritional composition of ingredients in the feed was assessed using Evapig® software [8], based on chemical analyses (dry matter, crude protein, starch, crude fibre, ash and fat contents, depending on the ingredient) performed by the feed company. Technical staff of the demonstration farm created a formulation to meet the nutritional recommendations (i.e., concentrations of net energy (NE) content, digestible lysine (and other essential amino acids), crude fibre, digestible phosphorus and calcium) at the lowest cost. The gestation diet was based on French guidelines [9] with two or three grains (wheat, barley and maize) and two or three types of meal (soya bean, rapeseed and sunflower), synthetic amino acids (lysine, methionine, threonine, tryptophan and valine when it was available at the feed plant), vegetable oil, sodium chloride, limestone, monocalcium phosphate and a vitamin and mineral mix. When energy concentration changed, it remained within a narrow range (9.0–9.4 MJ NE/kg), and the associated metabolizable energy (ME) concentration was calculated using the NE:ME ratio of ingredients published in feed tables [8]. The daily feed allowance was then obtained by dividing the daily ME requirement by the dietary ME concentration.

### 2.2. Measurements

Sows were weighed manually, and their BT was measured at the P2 position (at the last rib, 6.5 cm from the dorsal midline) using an ultrasonic linear probe (ECM, Angoulême, France). At the beginning of gestation, measurements were performed on the first Monday after insemination (i.e., ca. the 7th day of gestation). At the end of gestation, BT_final_ was measured on the last Monday before farrowing (ca. the 112th day of gestation), and sows were weighed within 24 h after farrowing (BW_final_). Piglets (born alive or stillborn) were weighed within the first 24 h of life (BiW_obs_) and the day before weaning (BW_w_, at 27 days of life), or when they died before weaning.

### 2.3. Modelling the Mean Daily Energy Requirement

The ME requirement, estimated using a factorial approach, equalled the sum of the daily maintenance requirement (ME_m_), daily requirement for physical activity (ME_act_) above the moderate level already considered in ME_m_, and the daily energy retention in maternal tissues (ME_ERs_) and in conceptus (ME_ERc_).

Data collected at the beginning of gestation were used to describe the initial characteristics of the sow (BW_initial_ and BT_initial_). The age at farrowing (Age_final_, day) was calculated as the sum of age at insemination (Age_initial_, day) and the mean duration of the gestation period in the herd (D_gest_, day). The relationship between BW_final_ and Age_final_ in the herd was used to calculate the expected net body weight of the sow at the end of gestation. The expected birthweight of the litter (BiW, kg) was determined from the BiW_obs_ of litters born from either primiparous or multiparous sows observed during the previous reproductive cycles of the farm. For cycles 20–26, BiW was set at 18.2 and 19.6 kg for primiparous and multiparous sows, respectively. It was increased respectively to 19.6 and 21.0 kg for cycles 26–44, 21.0 and 22.5 kg for cycles 45–47, and 21.8 and 23.2 kg for cycles 48–56. The weight of conceptus (BW_c_, Equation (1)) before farrowing included the placenta and fluids [10]: BWc (kg) = 0.3 + 1.329 BiW(1)

The ME_m_ (Equation (2)) and ME_act_ (Equation (3)) were considered proportional to the mean metabolic BW [11,12]:ME_m_ (MJ/d) = 0.44 BW^0.75^(2)
ME_act_ (MJ/d) = 0.27 BW^0.75^ (H × 60)/1000(3)
where BW^0.75^ equals the mean of BW_initial_^0.75^ and (BW_final_ + BW_c_)^0.75^, and H corresponds to the physical activity (hours) above the moderate level already assumed in ME_m_. 

Observations of sows in each type of housing system generated different assumptions. For sows housed in individual stalls or groups of 12 throughout the gestation period before housing reconstruction, 1 h of such physical activity was assumed, while for those housed in groups of 6, 2 h were assumed. After the reconstruction, 1 h was assumed for all sows when housed in stalls during the first month of gestation and during the last week after being moved to the farrowing room, while 2 h were assumed when group-housed in both systems (groups of 4–6 or 72). To represent the housing conditions throughout the gestation period, ME_act_ was assumed to be proportional to BW_initial_^0.75^, mean BW^0.75^ and (BW_final_ + BWc)^0.75^ sequentially.

The expected energy retention in maternal body reserves (ERr, Equation (5)) was calculated from the expected gain of empty body weight (ΔeBW, Equation (4)) and backfat thickness (ΔBT) during the gestation period [10]. The corresponding daily ME requirement was calculated (ME_ERs_, Equation (6)) with an efficiency of using ME for maternal retention of 77% [13]:ΔeBW (kg) = 0.905 BW_final_^1.013^ − 0.905 BW_initial_^1.013^(4)
ERs (MJ) = 13.65 ΔeBW + 45.94 ΔBT(5)
ME_ERs_ (MJ/d) = ERs/0.77/D_gest_(6)

The same target value for BT_final_ (20 mm) was used for all sows. This was empirically chosen when the farm was set up based on the expertise acquired previously from another herd with the same breed of sow. Investigations were then conducted to determine the most appropriate BT_final_ for the breed of sows on the farm. To this end, the values measured at the first farrowing were related to sow longevity in the herd (i.e., the number of farrowings before being culled).

The energy retention in conceptus (ER_c_, Equation (7)) was assumed to be proportional to BiW [6], and the corresponding mean daily ME requirement (ME_ERc_, Equation (8)) was calculated with an efficiency of using ME for conceptus growth of 48% [13,14]:ERc (MJ) = 5.44 BiW(7)
ME_ERc_ (MJ/d) = ERc/0.48/D_gest_(8)

The components of the daily requirement were quantified and divided by the dietary ME concentration (dME, MJ/kg) to obtain the mean daily amount of feed delivered (AFD, kg/d) to each sow (Equation (9)):AFD (kg/d) = (ME_m_ + ME_act_ + ME_ERs_ + ME_ERc_)/dME(9)

The AFD was multiplied by D_gest_ to calculate the total individual feed allowance per sow over the entire gestation period. From the day of the first insemination to the Wednesday of the following week (i.e., ca. the 9th day of gestation), the individual feed allowance was standardized to 2.7 kg/d for primiparous sows and 3.5, 3.2 or 2.8 kg/d for multiparous sows, depending on their BT at weaning (<16 mm, 16–20 mm, or >20 mm, respectively). From the 94th day of gestation to the day of farrowing, the individual feed allowance was standardized to 3.2 kg/d for primiparous sows and 3.5 kg/d for multiparous sows. For each sow, the cumulative amount of feed delivered before the 9th day and from the 94th day of gestation was calculated and subtracted from the total individual feed allowance. The difference was divided by the duration of the individualized feeding period to obtain the amount of feed to deliver to each sow based on her requirement.

### 2.4. Batches of Sows Studied and Calculations

Data from batches that farrowed in reproductive cycles 20–56 were considered. However, some batches were not included in the study when sows were fed specific diets, feeding strategies were followed in demonstration trials or data availability was restricted due to work performed in partnership with private companies. In addition, the COVID-19 crisis prevented measuring BT_final_ of sows or weighing piglets at birth in a few batches in 2020. On average, sows (*n* = 5140) from the 220 batches considered were fed 3.27 kg/d throughout the entire gestation period. The age at first farrowing averaged 375 days, with a standard deviation (SD) of 8 days, and the gestation period averaged 114.5 ± 1.3 days. Litter size averaged 15.6 ± 3.8 piglets (total born) at birth and 11.9 ± 1.8 at weaning, at ca. 27 days of age. The weight of the litter averaged 21.7 ± 4.7 kg at birth (BiW_obs_) and 101.1 ± 18.0 kg at weaning, with a daily gain of 3.07 ± 0.48 kg/d. The daily feed allowance averaged 3.27 ± 0.31 kg/d when the dietary NE content was 9.0 MJ/kg on an as-fed basis.

The literature describes a variety of models that were developed to describe the growth curves of many species [15,16,17,18,19]. The Gompertz function, for example, is difficult to calibrate when no information is available about the early stages of life. The first relationship used to predict BW_final_ was a Weibull model (Equation (10)) obtained from Large White sows [20], as it fitted the data collected during the first several reproductive cycles after the herd was established:BW_final_(Age_final_) = 273.7 × (1 − exp [−1.962/1000 × Age_final_^1.085^])(10)

Next, the relationship between BW_final_ and Age_final_ was modelled from sows that were studied for at least six parities over the reproductive cycles (SAS, v9.4, proc NLIN). Four sow populations were defined based on each sow’s year of birth, with one population defined every 4 years. Their root mean square error of prediction (RMSEP) was calculated for each relationship.

The mean and SD of measured BW_final_ and BT_final_ were calculated per batch that followed the precision feeding program and subjected to analysis of variance (SAS, v9.4, proc GLM), with the housing systems used before or after the station was under reconstruction as the main factor. Other investigations were based on individual data collected from sows from the nine batches that were not fed individually during the reconstruction period. Sows were classified based on BT_final_ ranges, and their performance at farrowing and during lactation was subjected to analysis of variance with the BT_final_ class as the fixed effect.

## 3. Results

### 3.1. Expected Final Backfat Thickness

#### 3.1.1. Definition of Optimal Backfat Thickness at Farrowing from Longevity

Data from 1221 primiparous sows born from February 2005 to April 2013 were collected from reproductive cycles 20–56 to study the relationship between BT_final_ at first parturition and longevity in the herd. When there were more gestating sows in the batch than places available in the farrowing unit, these sows were culled, and their data were removed from the analysis. Data from primiparous sows that farrowed during reproductive cycles 51–56 were also removed from the analysis because they could not reach 7th parity by that time. Ultimately, BT_final_ and the longevity of 720 primiparous sows were available, as was the housing system in which they were reared during the first gestation period. The mean longevity per rounded BT_final_ was calculated by housing system. Mean values obtained for at least 20 animals were calculated and plotted (Figure 1). Based on the first derivative of the polynomial equation obtained (Equation (11)), the maximum longevity corresponded to a BT_final_ of 21.6 mm at first farrowing.
Longevity = −0.0632 BT_final_^2^ + 2.7297 BT_final_ − 25.615 R^2^ = 0.8191(11)

#### 3.1.2. Definition of the Optimal Backfat Thickness at Farrowing Based on Performance

The sows from seven batches, chosen from the most heterogeneous batches studied when the farm was under reconstruction, were classed into five groups based on their BT_final_: ≤ 14, 15–17, 18–20, 21–23, and ≥24 mm. Differences in BT_final_ among sows from the same batch resulted from the sows competing for access to feeders. Sows in the 21–23 and ≥24 classes were significantly older and heavier at the beginning of gestation, and their BW and BT gains were also higher than those of sows in the other three classes (Table 1).

Despite having a similar parity (i.e., mean of 5.4), sows from class 21–23 were more prolific than sows from classes 18–20 and ≥24, with more total piglets born and weaned piglets per litter. The litters from class 21–23 tended to be heavier at birth, but when litter size was considered, litters from class 18–20 had the highest mean birthweight per piglet. The class of BT_final_ had no significant effect on the daily gain of the litter, even though it was 0.25 kg/d higher in class 21–23 than in class ≤14.

### 3.2. Expected Final Body Weight 

Four populations of sows were considered to model relationship between BW_final_ and Age_final_. The mean and standard deviation of Age_final_ and BW_final_ per parity and populations varied by class (Table 2).

The equations obtained using a monomolecular Brody model (Equations (12)–(15)) or a generalized Weibull model (Equations (16)–(19)) varied by population, as did their RMSEP. Mature BW_final_ varied little among populations, and the Brody and Weibull models each obtained a narrow and similar range of values (308–331 and 310–335 kg, respectively). The difference between individual BW_final_ for each parity of the 90 sows in the population born from 2012–2015 and the BW_final_ predicted with the Weibull model was plotted (Figure 2). The RMSEP per population ranged from 13.1–16.1 kg using the Brody model and from 12.1–14.6 kg using the Weibull model. 

Modelling BW_final_ as a function of Age_final_ using the monomolecular Brody model:Population 1: BW_final_ (Age_final_) = 325.6 × (1 − 0.800 × exp(−2.170/1000 × Age_final_)) RMSEP = 14.6(12)


Population 2: BW_final_ (Age_final_) = 308.5 × (1 − 0.841 × exp(−2.664/1000 × Age_final_)) RMSEP = 13.1(13)



Population 3: BW_final_ (Age_final_) = 315.0 × (1 − 0.818 × exp(−2.589/1000 × Age_final_)) RMSEP = 14.1(14)



Population 4: BW_final_ (Age_final_) = 331.4 × (1 − 0.821 × exp(−2.121/1000 × Age_final_)) RMSEP = 16.1(15)


Modelling BW_final_ as a function of Age_final_ using the generalized Weibull model:Population 1: BW_final_ (Age_final_) = 338.0 (1 − exp [−(2.577/1000 × Age_final_)^0.773^]) RMSEP = 14.4(16)


Population 2: BW_final_ (Age_final_) = 310.0 (1 − exp [−(3.164/1000 × Age_final_)^0.871^]) RMSEP = 12.1(17)



Population 3: BW_final_ (Age_final_) = 318.2 (1 − exp [−(3.147/1000 × Age_final_)^0.845^]) RMSEP = 13.3(18)



Population 4: BW_final_ (Age_final_) = 334.6 (1 − exp [−(2.634/1000 × Age_final_)^0.832^]) RMSEP = 14.6(19)


When the difference between the measured and predicted BW_final_ at each farrowing was calculated for all sows in each population (not only for those that farrowed at least six times), the RMSEP of the Brody model ranged from 12.2–17.5 kg, and that of the Weibull model ranged from 12.3–17.3 kg (i.e., similar to the RMSEP obtained only from data used to calibrate the models). In contrast, the original Weibull model (Equation (10)) obtained a higher RMSEP, which ranged from 25.8–42.7 kg depending on the population. The RMSEP decreased (12.4–21.9 kg) when the mature BW_final_ was changed to 300.0 kg.

### 3.3. Expected and Observed Birthweight of the Litter

The mean BiW_obs_ of litters farrowed by primiparous or multiparous sows in the 220 batches considered in the present paper is illustrated in Figure 3. Litters born from primiparous sows weighed a mean of 19.3 kg, with a SD of 2.0 kg among batches. The corresponding mean BiW_obs_ for litters born from multiparous sows was 22.4 kg, with a SD of 1.5 kg. Over the 220 successive batches (N), BiW_obs_ and N had a stronger correlation in litters born from multiparous sows (r = 0.54, *p* < 0.001) than in those born from primiparous sows (r = 0.19, *p* = 0.004). Linear regression of BiWobs as a function of N indicated that BiW_obs_ increased by a mean of 13 ± 1 g/batch when born from multiparous sows (BiW_obs_ = 20.957 (±0.171) + 0.012 (±0.001) N, R^2^ = 0.29). Only 4% of the variability in BiW_obs_ litters born from primiparous sows was related to changes in BiW_obs_ among batches (BiW_obs_ = 18.596 (±0.267) + 0.006 (±0.002) N, R^2^ = 0.04).

### 3.4. Mean and within-Batch Variability in Final Characteristics

The mean BW_final_ and within-batch SD obtained from 220 batches varied (Figure 4a,b), as did the corresponding BT_final_ (Figure 4c,d). The transition period included nine batches (numbers 119–128 in Figure 4). Their mean BW_final_ (261.3 ± 41.2 kg) and BT_final_ (20.2 ± 4.8 mm) were similar to those obtained in systems in which sows were fed individually (Table 3). In contrast, some of the transitional batches had the highest SD for BW_final_ and BT_final_ (Figure 4b,c), which resulted from greater competition for feed among sows kept in large pens in the fattening unit.

Mean BW_final_ did not differ significantly among batches housed in different housing systems used before reconstructing the gestation unit (*p* = 0.19, Table 3); however, the within-batch variability in mean BW_final_ was higher in groups of 12 sows than in the other groups (36.4 vs. 32.6 kg, *p* < 0.001). After the reconstruction, the two housing systems had similar mean BW_final_ (*p* = 0.75). Sows housed in small groups were significantly heavier than before (*p* = 0.003), but the difference in means was only 4 kg (268.4 vs. 262.2 kg, respectively). As noted, the variability in BW_final_ per batch was consistently higher when sows were housed in a large group than when they were housed in a small group (Figure 4b), and the difference in means was significant (*p* < 0.001, Table 3). 

Before the reconstruction, mean BT_final_ was significantly higher (by 0.6 mm) in batches of sows housed in stalls than in groups of 6 or 12 (18.7 vs. 19.3 mm, respectively, *p* = 0.05). After the reconstruction, mean BT_final_ tended to be 0.6 mm higher for batches of sows housed in groups of 72 than in groups of 4–6 (19.7 vs. 19.1, respectively, *p* = 0.07). The mean (*p* = 0.55) and SD (*p* = 0.27) of BT_final_ did not differ significantly between batches housed in small groups before vs. after reconstruction. The SD of BT_final_ averaged 2.9 mm in before reconstruction and 3.1 mm afterwards, and it tended to be higher in large groups.

## 4. Discussion

The equations based on the four populations recorded every four years indicate that it is not necessary to update the equation used to predict BW_final_ every four years; it can be updated less frequently. The Brody and Weibull models fit the data similarly. The precise calibration of the sow profiles is based on the dataset used to describe mature BW. The original Weibull equation (Equation (10)) initially fit the data obtained from the first reproductive cycles, but it seemed that it was evaluated too soon. After only a few reproductive cycles, the amount of available data from old sows was too limited, which resulted in underestimating the mature BW_final_ (mean of 273.4 kg from Equation (10), compared to 320.1 kg with the Brody model and 325.2 kg with the Weibull model). Because the mean BW was slightly underestimated, so was the energy requirement for maintenance and physical activity. This could result in a difference between expected and observed BT_final_.

The optimum BT_final_ can be characterized in different ways. Comparing results that have different ranges of values requires time-consuming measurements of many sows so that significant differences can be analysed. A more practical solution would be to consistently measure the BT of all primiparous sows at farrowing and to relate them to longevity in the herd. Sows are culled from the herd for many reasons, such as excessive development in young animals (lameness) or excessive nutritional imbalance and use of body reserves during lactation (reproduction). Both strategies provide a similar expected BT_final_ (21 mm) for the breed of sows on the farm. This target value is specific to the breed of sow and the characteristics of the farm, such as the feeding system, housing conditions and management. A different target value could be obtained on another farm for the same breed of sow, even when following the same feeding strategy. Additional criteria could be considered to select the target BT_final_ in order to find a compromise among the impacts of BT on the components of sow performance. Data collected from individual sows on two experimental farms reveal a linear increase in piglet birthweight and the pre-weaning growth rate of the litter as the individual BT_final_ increased; however, the number of piglets born alive per litter and the appetite of the sow decreased [21]. On commercial farms, these data are frequently unavailable or stored in different ways (e.g., herd books, separate files). In the future, using sensors and equipment to provide phenotypes of animals and facilitate data interoperability will provide many opportunities for smart farming, including determining the BT_final_.

The similar mean BW_final_ and BT_final_ obtained across housing systems could indicate that the assumptions about the average level of physical activity used to model the ME_act_ requirement of sows as a function of the housing system could be accurate, or at least not too far from reality. In contrast, the difference in SD among housing systems could indicate this was not the case for each individual. No information was available about inter-individual variability in the number of hours of physical activity in different postures; therefore, it could not be considered, and the average assumption used for all sows in a given housing system might differ greatly from reality at the individual level. This may have influenced the within-batch variability in BT_final_. The significantly higher SD of BW_final_ and the trend for a higher SD of BT_final_ for groups of 12 or 72 compared to those for other housing systems (small pens or stalls) is consistent with the increased variability in motion when the size of the pen associated with a larger group increased [22]. The variability in effective ME_act_ requirement resulted in more or less ME available for production, especially for energy retention in maternal tissues. In addition, ME_act_ was estimated based on the influence of the duration of standing on heat production [13]. However, walking or running may expend more energy than standing [23].

From October 2005 to August 2020 (i.e., the insemination period of the 220 batches), the expected value of BiW used to calculate the ME_ERc_ requirement increased from 18.2 to 21.8 kg for piglets born from primiparous sows, and from 19.6 to 23.2 kg for piglets born from multiparous sows. The increase in the expected BiW was slightly higher than that in BiW_obs_ for primiparous sows. As it is not currently possible to characterize foetal development in utero, it appears necessary to consider the progress observed in litter size on the farm and to anticipate the ongoing progress in genetic selection. Otherwise, this might have resulted in underestimating ME_ERc_ at the expense of the ME assumed to be available for energy retention in maternal tissue. Moreover, the increase in expected BiW did not reflect the large variation in BiW_obs_ among batches (Figure 3) and among sows (data not shown), which also influenced the variability in BT_final_ among batches (Figure 4c) and within batches (Figure 4d). Finally, when some sows have a BT_initial_ that is already higher than the target BT_final_, it becomes more difficult to attain the expected BT_final_ for all sows. In this situation, the aim of the feeding strategy is not to make the sow thinner at farrowing than it is at insemination, when energy supplies are calculated, assuming no energy retention in the maternal body.

## 5. Conclusions and Perspectives

The precision feeding strategy implemented, with a single gestation diet, helps to monitor the body condition of sows at farrowing using energy supplies that are adapted to energy requirements. Individual ME requirements were estimated by measuring initial BW, initial BT and age, and using common expected values for final BW (depending on age), final BT, physical activity level (depending on the housing system) and litter birthweight (depending on parity). Precision feeding decreased the risk of high within-batch variability at farrowing compared to that when feed allowance is not adapted to meet requirements. Thus, it also decreased the risk of having sows that were too fat or too thin, which decreased the risk of impairing farrowing or milk production, and improved animal welfare (for sows and their progeny). However, greater accuracy in estimating energy supplies is expected in the near future.

The knowledge available about nutritional requirements of sows during different periods of the reproductive cycle is aggregated in models that can be incorporated in decision support systems related to the feeding system and tools used to characterize the sows on each farm. Sensors and automation are continuing to develop on farms and are becoming less expensive. These tools provide new opportunities to consider dynamic changes in individual body weight (with automatic weighing scales) and individual physical activity (e.g., with video recording or accelerometers). However, new tools are expected to characterize the composition of body weight gain or in utero foetal growth. This kind of new information, collected daily and individually, will be useful for assessing requirements besides energy, such as amino acids and minerals. Electronic feed dispensers equipped with two or three hoppers will enable blending different diets, formulated for different nutrient levels, so that the quality of the diet can be individualized, not only the quantity. 

To date, addressing variability in sow characteristics has largely been the responsibility of farmers. A precision-feeding system could make it simpler to adapt nutrient supplies to the body reserves of sows and to monitor body condition at farrowing. Precision feeding could also improve nutrient-use efficiency and decrease environmental impacts and feed costs.

## Figures and Tables

**Figure 1 animals-11-02908-f001:**
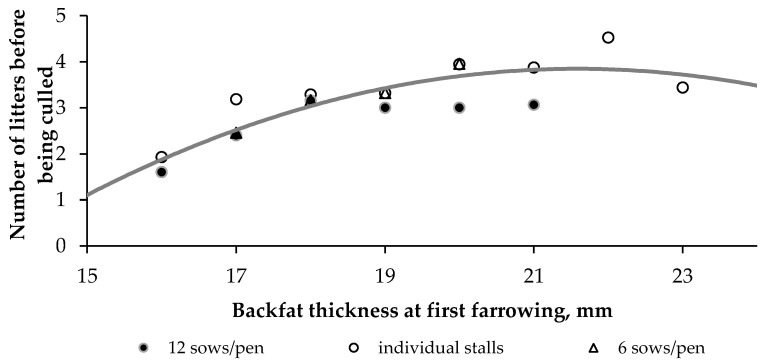
Relationship between mean backfat thickness measured at first farrowing and mean longevity (i.e., number of litters farrowed before being culled from the herd) for sows housed in individual stalls or in pens containing 6 or 12 sows (from cycles 20–39, 720 primiparous sows, each dot represents at least 20 sows).

**Figure 2 animals-11-02908-f002:**
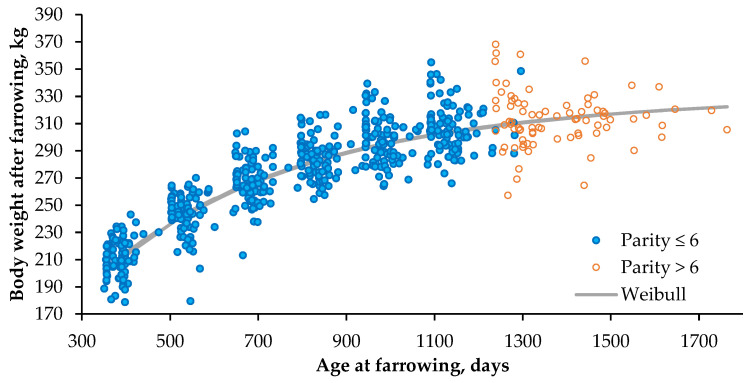
Individual ages and body weights at farrowing of sows in population 4 (born from 2012–2015), along with body weight predicted using the Weibull model (Equation (19)).

**Figure 3 animals-11-02908-f003:**
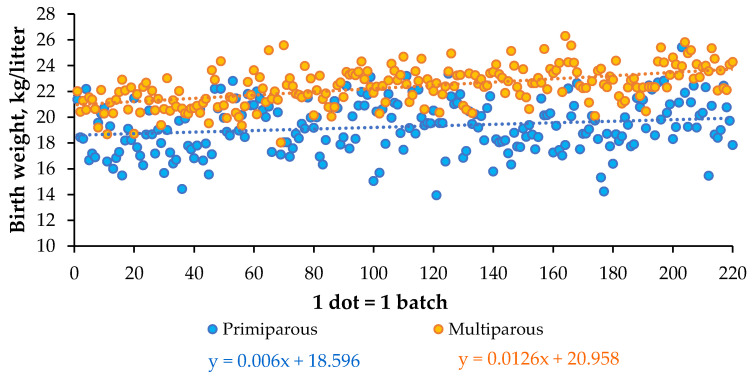
Mean birthweight (kg/L) obtained per batch from primiparous and multiparous sows in 220 batches from reproductive cycles 20–56.

**Figure 4 animals-11-02908-f004:**
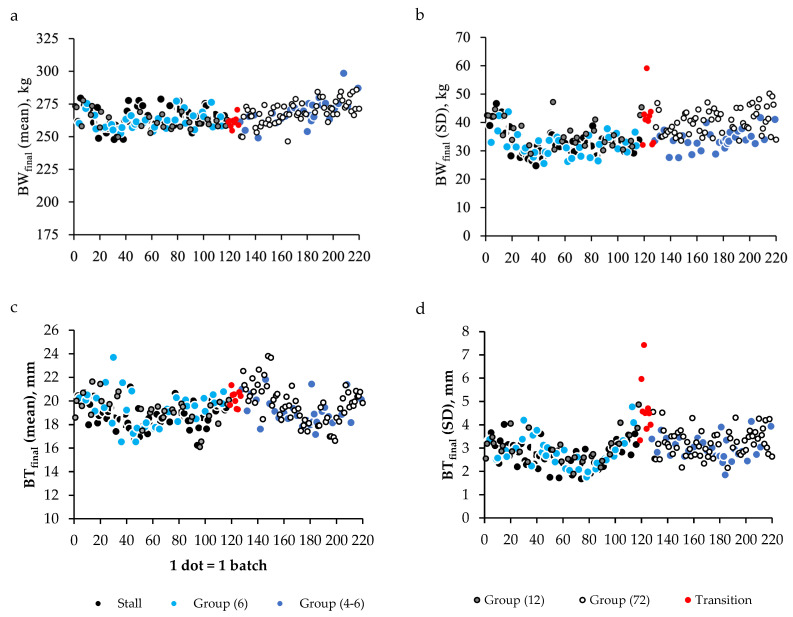
Mean and standard deviation (SD) of body weight after farrowing (BW_final_, (**a**,**b**)) and backfat thickness at farrowing (BT_final_, (**c**,**d**)) calculated per batch from cycles 20–56 (220 batches), housed in different systems: individual stalls, small groups (6 sows per pen) or large groups (12 sows per pen) for the entire gestation period for the first 118 batches; in a transition system (without individualized feed allowance) when the farm was under reconstruction for 9 batches; and, small groups (4–6 sows per pen) or large groups (up to 72 sows per pen from 3 batches housed simultaneously) for the group-housed phase of the gestation period for the last 93 batches.

**Table 1 animals-11-02908-t001:** Mean performance at birth and during lactation by class of backfat thickness at farrowing (BT_final_).

						Stat.	
BT_final_ Class, mm	≤14	15–17	18–20	21–23	≥24	RSD ^1^	*p*-Value ^2^
Number of sows	15	23	21	19	23		
Parity	4.0	4.0	5.2	5.7	5.3	2.0	0.03
Body weight, kg							
At 7 d of gestation	216	215	231	241	242	26	<0.01
At farrowing	232	256	279	295	310	28	<0.01
At weaning	223	236	258	265	274	27	<0.01
Backfat thickness, mm							
At 7 d of gestation	13.7	15.4	15.3	16.0	17.1	2.7	<0.01
At farrowing	11.1	16.2	19.0	22.1	27.6	2.2	<0.01
At weaning	10.1	13.3	15.1	16.7	19.7	2.0	<0.01
Litter size							
Total piglets born	17.1	15.0	15.1	17.8	16.2	3.5	0.05
Piglets born alive	16.3	14.3	14.1	16.5	15.0	3.3	0.06
Weaned	11.9	12.0	11.6	12.2	11.6	1.6	0.65
Litter weight, kg							
At birth	21.8	21.6	22.8	25.1	23.3	3.0	0.07
At 27 d of age	97.0	103.0	105.6	105.0	99.5	13.8	0.44
Piglet birthweight, kg							
At birth	1.32	1.50	1.56	1.42	1.47	0.23	0.02
At 27 d of age	8.14	8.62	9.15	8.67	8.68	0.81	<0.01
Litter growth rate, kg/d	2.98	3.18	3.20	3.23	3.10	0.38	0.46

^1^ RSD: relative standard deviation. ^2^ Analysis of variance (SAS, v9.4, proc GLM) with BT_final_ and the batch as the main effects, and litter size as a covariate for litter weight at birth or at weaning at 27 days of age.

**Table 2 animals-11-02908-t002:** Mean ± standard deviation of age (Age_fianl_, d) and body weight (BW_final_, kg) at farrowing in Large White × Landrace sows studied from parity 1 to 6, depending on the population in the herd (i.e., year of birth).

Parity	1	2	3	4	5	6
Population 1: sows born from 2000–2003 (116 sows)						
Age_final_	380.9 ± 12.6	520.6 ± 15.9	667.8 ± 16.7	817.3 ± 20.3	966.3 ± 21.4	1118.0 ± 26.3
BW_final_	209.5 ± 10.7	241.2 ± 14.7	265.1 ± 15.3	279.9 ± 14.1	293.4 ± 14.1	303.2 ± 15.1
Population 2: sows born from 2004–2007 (109 sows)						
Age_final_	374.8 ± 12.7	525.7 ± 18.4	674.2 ± 19.3	822.5 ± 19.8	871.9 ± 22.2	1121.6 ± 24.3
BW_final_	212.8 ± 9.1	244.5 ± 10.0	266.7 ± 11.2	278.8 ± 11.4	289.0 ± 12.9	295.6 ± 13.2
Population 3: sows born from 2008–2011 (112 sows)						
Age_final_	379.0 ± 14.7	531.4 ± 21.1	680.9 ± 24.5	830.4 ± 29.1	983.0 ± 37.1	1131.1 ± 39.0
BW_final_	218.2 ± 10.0	250.8 ± 10.7	269.4 ± 12.9	286.2 ± 14.0	295.5 ± 14.7	300.7 ± 13.7
Population 4: sows born from 2012–2015 (90 sows)						
Age_final_	383.8 ± 22.4	539.2 ± 34.7	690.5 ± 40.9	840.3 ± 43.0	993.3 ± 47.3	1141.1 ± 47.5
BW_final_	211.3 ± 13.3	243.3 ± 14.6	268.4 ± 15.1	285.5 ± 13.8	298.7 ± 15.9	306.8 ± 17.2

**Table 3 animals-11-02908-t003:** Overall mean and standard deviation (SD) of body weight (BW_final_) and backfat thickness (BT_final_) at farrowing calculated per batch in different housing systems before and after the reconstruction period on the farm.

		BW_final_, kg		BT_final_, kg	
Housing Systems	*n*	Mean	SD	Mean	SD
Before reconstruction					
12 sows/pen	32	261.9	36.4 ^a^	19.3 ^a^	3.0
6 sows/pen	35	262.2	32.3 ^b^	19.3 ^a^	2.9
Individual stall	51	264.6	32.9 ^b^	18.7 ^b^	2.7
Statistics					
RSD ^1^		7.4	4.6	1.2	0.6
*p*-Value ^2^		0.19	<0.001	0.05	0.07
After reconstruction					
72 sows/pen	63	269.1	40.3 ^A^	19.7	3.2
4 or 6 sows/pen	30	268.4	34.0 ^B^	19.1	3.0
Statistics					
RSD ^1^		9.0	4.1	1.5	0.6
*p*-Value ^2^		0.75	<0.001	0.07	0.08

^1^ RSD: residual standard deviation. ^2^ Analysis of variance performed on mean body weight (BW_final_) and backfat thickness at farrowing (BT_final_) per batch and within-batch standard deviation, with the housing system as the main effect. Different superscripts (a and b before reconstruction; A and B after reconstruction) indicate that values in rows differ significantly (*p*-Value ≤ 0.05).

## Data Availability

Data were obtained from the SCEA du Grand Clos (35850 Romillé, France) and are available from the author with the permission of the SCEA du Grand Clos.

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
