# Peer review of "Results of 15 Years of Precision Feeding of Hyper Prolific Gestating Sows"

_animals, 2021, doi:10.3390/ani11102908_

Round 1

Reviewer 1 Report

The manuscript presents the descriptive statistics of gestating sows that were fed a tailored amount of energy allowance. The tailored amount of feed was calculated from models based on age, body weight, and backfat at the beginning of gestation. The consistency of the work over the 25 years deserves a commendation.

The key question I was hoping to ask is whether the farm that adopted precision feeding has achieved an improved uniformity in sow body conditions, birth weight, or litter size? As a standard (or expected) pre-farrowing body condition and average litter standard weight were used for developing the energy allowance, I guess the answer is yes.  Wish the author could provide some evidence.

Additionally, could you please comment or provide some figures on whether precision feeding increases farm profitability? Does the feeding-to-expected energy allowance constrain the productivity of high-performance sows (eg: the same final backfat target is used)? It would be important feedback.

A question on the procedure to set BWFinal. As stated in the method section, BWFinal was initially set based on the published relationship between age and BWfinal (Equation 10). Then a new relationship was established when data points from the farm had become available. Had the subsequent new relationship been dictated (or pre-determined) by the published relationship used before? As the sows had been fed to achieve Equation 10-expected final bw, then the bw was used to determine the new relationship between age and final bw.

Should an optimal BWfinal (i.e.: not only age-dependent but also can achieve the best reproductive performance or longevity) be modelled from the data and then used for setting the BWFinal?

Need more information in the method section on the genetic selection for the four generations. How was the progeny selected for each new generation? What maternal traits were used for genetic selection? What was the proportion of the progeny got selected into the breeder herd?

L77-84 Was the same dietary specification used for all the batches?

L87 Please describe the probe manufacturer

L103-105 How did you set the expected birth weight for a pregnant gilt?

L103-105 The “mean BiWobs” may confuse. Did you use the average birth weight pooled across the previous parities of each sow or the average birth weight of the population of primiparous/ multiparous sows? Please clarify.

L131 Why was 20 mm chosen as the backfat target at farrowing for all the sows? Can producers from other countries follow the same backfat target? Should it be different by parities?

Equation 5 “ERs” or “ERr”

Equation 4 why empty maternal body weight for initial and end of gestational was calculated using the same term (0.905 BW1.013)? Should the sows at the end of gestation have a greater proportion of conceptus than the early gestation?

L146 I am a bit confused by “the reproductive cycles 20 to 56”. Do you specifically mean the oestrus cycle? Maybe I have a different understanding of the phrase “reproductive cycle”- I think one reproductive cycle of the sow consists of gestation, lactation, and a weaning-to-remating interval.

L151-152 Was the feed level different by the phase of gestation? Please describe in detail.

L157-158 Can you please report the mean and standard deviation of the calculated metabolizable energy (ME) requirement and longevity?

In the method section, can you please describe the culling criteria used in this farm? Because it is relevant to longevity.

L166-168 Why was the population considered to be different every 4 years?

Table 1 number of sows in each backfat category is very small. Why was such a small number of sows were selected for the analysis in Table 1?

Table 1 The sows in <14 mm group gained bodyweight but reduced backfat (11.1 mm to 10.1 mm) from d7 gestation to farrowing. Was it due to the small sample size used? Please double-check the data.

L319 Should a farm update the equations to predict BWfinal more frequently if it does rigorous genetic selection on backfat, litter size, and average birth weight in the breeding program?

Author Response

Rev1. comments:*
The manuscript presents the descriptive statistics of gestating sows that were fed a tailored amount of
energy allowance. The tailored amount of feed was calculated from models based on age, body
weight, and backfat at the beginning of gestation. The consistency of the work over the 25 years
deserves a commendation.
The key question I was hoping to ask is whether the farm that adopted precision feeding has achieved
an improved uniformity in sow body conditions, birth weight, or litter size? As a standard (or expected)
pre-farrowing body condition and average litter standard weight were used for developing the energy
allowance, I guess the answer is yes. Wish the author could provide some evidence.
Answer: The data obtained within the farm cannot be used to answer to this crucial question. Another
paper in the special Issue will perhaps demonstrates this. Of course, these are important aims of the
feeding strategy, and it is the reason why it was used. When the demonstration farm was built, farmers
referred to the available literature on the importance to be paid to the final backfat thickness at
farrowing, its connection with feed allowance and the impact of the within-batch variability on herd
performance. The improvement of sow body condition and a decrease in variability at farrowing was
the main objective on a short-term basis. Variability in litter size of birth weight needs to be considered
on a longer-term basis, as the performance during lactation (energy unbalance) and at the beginning
of gestation (weaning to estrus interval) interferes.
In the demonstration farm, all sows were managed with the precision feeding system, and no control
group was available to evaluate the impacts of precision feeding compared to a standard strategy, with
the standard being not the same today like 25 years ago. However, an indirect answer can be given
when we compare the performance of the demonstration herd to others with the same type of sow. In
the demonstration farm, it is possible to achieve a high final backfat with a low standard deviation.
Then, we can manage sows with an average energy store which allows to produce a lot of milk over 4
weeks, but without too many extreme sows that would present an increased risk of problems at
farrowing due to this high target value if it was associated with a high SD. Of course, there are still too
fat sows in the herd (who contribute to the SD of final backfat). But they are fat at farrowing because
they were already fat at insemination and the aim of the feeding strategy is not to make them loose BT
in that case as mentioned in the paper. Then, with a target BF of 20 +/- 3 mm, sows can wean litters of
12 piglets that weigh 101 kg. In commercial farms, with a target BF of 16 to 18 mm, litters are lighter at
birth.
Additionally, could you please comment or provide some figures on whether precision feeding
increases farm profitability?
Another paper in the special Issue will perhaps demonstrates this. Kansas State university indicates
on the web that a feeding program based on BW and BF would help to reduce feed costs by 10 USD
(Feeding the Gestating Sow – Hogs, Pigs, and Pork (extension.org). Cloutier et al. (2019) focused on
the improved adequacy between amino acid supply and requirements (http://www.journees-rechercheporcine.
com/texte/2019/alimentation/a18.pdf) was found that the feed costs during gestation would be
reduced by 2 €. Other costs are not considered here, corresponding the following hypotheses: saving
labor cost because less supervision is required and increased survival rate of piglets when difficult
farrowing are avoided with less too fat sows, (ii) improved longevity of sows, (iii) better milk production
with less too thin sows associated with heavier weaning weight. But this is not the scope of the paper,
and it is not possible to evaluate this from the comparison with a standard strategy as indicated above.
Does the feeding-to-expected energy allowance constrain the productivity of high-performance sows
(eg: the same final backfat target is used)? It would be important feedback.
Recent results indicate that milk production in the farm is not limited by precision feeding, as both litter
weight (111 kg) and size (13.7 piglets) at weaning keep improving.
Compared to the average prolificacy observed in French commercial farms, which average results are
collected each year, the litter size obtained in the demonstration farm is 1 piglet higher on average.
The annual progress is + 0.20 additional piglet/litter in the demonstration farm, vs. +0.15 in commercial
farms. The difference is not explained only by the feeding strategy, but it indicates at least that the
precision feeding does not limit the genetic progress observed elsewhere. The figure below illustrates
the average annual litter size (total born) observed at the national level and in the demonstration farm.
This figure is confidential.
A question on the procedure to set BWFinal. As stated in the method section, BWFinal was initially set
based on the published relationship between age and BWfinal (Equation 10). Then a new relationship
was established when data points from the farm had become available. Had the subsequent new
relationship been dictated (or pre-determined) by the published relationship used before? As the sows
had been fed to achieve Equation 10-expected final bw, then the bw was used to determine the new
relationship between age and final bw.
Answer: Yes, it is true that the data used to calibrate the growth profile of sows were obtained under
energy supplies supposed to fit another growth curve. Energy supplies for maternal energy retention
are calculated from the difference between the initial and the final amounts of body energy, supposed
to correspond to a BW and BF gain over the period. But in practice the BW gain was often higher than
the expectation. Even when the energy allowance was based on a mature BW of 274 kg, sows were
able to reach a mature BW that was much higher. It would agree with the concept that considers the
protein retention as the driving force of growth, like during the growing-finishing period. It was possible
without a huge impact on final backfat because of the low energy cost of the BW gain associated to
extra protein deposition (compared to lipid), and because amino acid supplies were not limiting as only
one diet was used during the whole gestation, which concentration in amino acids was chosen to meet
the requirement of young sows at the end of the gestation. In addition, the energy requirement for
maternal gain represents only around 20% of the total energy requirement (to add to maintenance at
75% and fetal growth 5%). Anyway, the difference between the observed final backfat and the target
could be explained by the extra BW gain (above the expectation) that resulted in more important
maintenance requirement and less energy available for lipid retention.
When precision feeding accounts for components of the maintenance requirement more precisely
(especially with regard to individual daily physical activity and body weight) and for the characteristics
of the litter, a discrepancy between the objective and the real BW gain may have more important
consequences on the final BF.
Should an optimal BWfinal (i.e.: not only age-dependent but also can achieve the best reproductive
performance or longevity) be modelled from the data and then used for setting the BWFinal?
Answer: From a conceptual point of view, probably yes. It is more in the scope of what can be done
with results like those published by Lavery et al. (2019) who made association between parity, liveweight
and backfat and productivity. Such investigations require a large data set.
Need more information in the method section on the genetic selection for the four generations. How
was the progeny selected for each new generation? What maternal traits were used for genetic
selection? What was the proportion of the progeny got selected into the breeder herd?
Answer: There is a misunderstanding. The demonstration farms buy the replacement gilts from a
commercial multiplication herd, in which animals are produced from dam that come from a nucleus
herd. In the paper, what we call “generation” corresponds to successive deliveries of replacement gilts
that have been gathered by their year of birth. Gilts from one group were not farrowed by those of the
previous one. The text has been modified: generation has been replaced by population everywhere.
L77-84 Was the same dietary specification used for all the batches?
Answer: sentences were added L86-L95
The gestation diet was formulated based on recommendations of IFIP et al. (2002) with two or three
grains (wheat, barley, corn) and two or three meals (soybean, rapeseed and sunflower), synthetic
amino acid (lysine, methionine, threonine, tryptophane and valine when it became available at the
feed plant), vegetal oil, sodium chloride, limestone, monocalcium phosphate and vitamin and mineral
mix. When changes in energy concentrations occurred, it remained within a narrow range of value (9.0
to 9.4 MJ NE/kg), and the associated metabolizable energy (ME) concentration was calculated with
reference to NE/ME ratio of feedstuffs published in tables (INRA, AFZ, CIRAD, 2004). Thereafter, the
daily allowance of feed was obtained by dividing the daily ME requirement by the dietary ME
concentration.
L87 Please describe the probe manufacturer
L98 – added (ECM, Angoulême, France)
L103-105 How did you set the expected birth weight for a pregnant gilt?
L103-105 The “mean BiWobs” may confuse. Did you use the average birth weight pooled across the
previous parities of each sow or the average birth weight of the population of primiparous/ multiparous
sows? Please clarify.
Answer L114 –119 More details are given on the hypotheses used for expected the birth weight.
The expected birthweight of the litter (BiW, kg) was decided from the BiWobs of litters born either by
primiparous or multiparous sows observed during the previous reproductive cycles of the farm. For
cycles 20 to 26, the value was fixed to 18.2 and 19.6 kg for primiparous and multiparous sows,
respectively. It was increased to 19.6 and 21.0 kg for cycles 26 to 44, to 21.0 and 22.5 kg for cycles 45
to 47, and 21.8 and 23.2 kg for cycles 48 to 56.
L424-431 Sentences have been modified accordingly.
From October 2005 to August 2020, i.e. the insemination period of the 220 batches, the expected
values of BiW used to calculate the MEERc requirement increased from 18.2 to 21.8 kg when born
from primiparous sows, and from 19.6 to 23.2 kg when born from multipa-rous sows. The increase in
target BiW was close to but slightly higher than the increase observed in BiWobs. As long as it is not
possible to characterize in utero the development of fetuses, it appears necessary to account for the
progress observed in litter size on farm and to anticipate the ongoing progress of genetic selection.
Otherwise, it might have resulted in an underestimation of MEERc,
L131 Why was 20 mm chosen as the backfat target at farrowing for all the sows? Can producers from
other countries follow the same backfat target? Should it be different by parities?
Answer:
The target of 20 mm was chosen relying on the expertise acquired on the same breed of sows in
another farm as indicated presently L131-132, which was equipped with similar feeding system in
lactation rooms.
It was chosen at the start-up of the farm empirically, relying on the expertise acquired pre-viously in
another herd with the same type of sow.
We agree that it is not a universal target, and that many criteria have to be considered. It should be
adapted to numerous factors. The type of sow influences the performance level in terms of prolificacy
(i.e., requirements for fetal growth during late gestation and/or for milk production after birth) and in
terms of appetite during lactation. Then, the target can be reduced for sows with a lower prolificacy
(lower milk production) or with a higher appetite (associated with a better energy balance during
lactation). However, for a given type of sow, the feeding system is important to consider. During
gestation, when it is possible to individualize the amount of feed delivered it is easier to achieve a high
value for the target BF without too extreme values in BF (especially avoiding too fat sows because of
impaired farrowing process) than when it is not possible. In such conditions, when the standard
deviation of BF among sows of the batch is rather high and cannot be reduced, the farmers usually
decide to decrease the target BF. Thereafter, the herd is thinner on average at farrowing and the risk
for a sow to be too fat is reduced. But it that condition, the amount of body energy can be too low to
compensate for a negative energy balance and the litter’s growth rate (e.g., milk production) is
reduced as also observed by Lavery et al. (2019) (reference number 20 added in the list).
New sentences have been added L372-384: This target value is per se specific to the type of sow and
the characteristics of the farm in terms of feeding system, housing conditions, management… For the
same type of sow, a different value may be obtained in another farm, even when the same feeding
strategy is implemented. Additional criteria may be considered to choose the target BTfinal, seen as a
compromise between the impacts of BT on the different components of sows’ performance. Based on
data collected on individual sows in two experimental farms, a linear increase in piglets’ birth weight
and in pre-weaning growth rate of the litter was observed when the individual BTfinal increased, but
the number of born alive piglets per litter and the appetite of the sow decreased [20]. In commercial
farms, these data are presently most often unavailable or stored in different ways (herd books,
separate files…). But in a close future, the utilization of sensors and any equipment that contribute to
phenotype the animals and facilitate the data inter-operability will offer many opportunities for smart
farming, including the definition of the BTfinal for example.
Equation 4 why empty maternal body weight for initial and end of gestational was calculated using the
same term (0.905 BW1.013)? Should the sows at the end of gestation have a greater proportion of
conceptus than the early gestation?
Answer: The empty body weight does not correspond to body weight minus conceptus, but to body
weight minus digestive tract contents.
L146 I am a bit confused by “the reproductive cycles 20 to 56”. Do you specifically mean the oestrus
cycle? Maybe I have a different understanding of the phrase “reproductive cycle”- I think one
reproductive cycle of the sow consists of gestation, lactation, and a weaning-to-remating interval.
Answer: L55-57 detail is given to explain the reproductive cycle considered later
“After weaning, the number given to the batch n (with n ranging between 1 and 7) is incremented to
account for the number of times the batch was reproduced (so called reproductive cycle of the farm)”.
L151-152 Was the feed level different by the phase of gestation? Please describe in detail.
Answer: sentences added L159-170 to detail our the individual total amount of feed was deliver, with
an individual plan between the 9th and the 94th day of gestation.
The AFD was multiplied by Dgest to calculate the total individual feed allowance per sow over the
whole gestation. However, it was not possible to individualize the feed allowance from the
insemination to farrowing. From the day of the first insemination to the Wednesday of the following
week, the feeding plan corresponded to 2.47 kg/d for primiparous sows and either 3.5, 3.2 or 2.8 kg/d
for multiparous sows, depending on their BT at weaning (below 16 mm, between 16 and 20 mm, or
above 20 mm, respectively). Between the 94th day of gestation and the day of farrowing, the feeding
plan was standardized to 3.2 kg/d for all primiparous sows and to 3.5 kg/d for all multiparous sows 3.5
kg/d. Then, for each sow, the cumulated amount of feed delivered before the 9th day and to be
delivered at the end of the gestation was calculated and subtracted from its total individual feed
allowance. The difference was divided by the duration of the individualized feeding period to obtain the
amount of feed to deliver to each sow based on her requirement.
L157-158 Can you please report the mean and standard deviation of the calculated metabolizable
energy (ME) requirement and longevity?
Answer: From 1 802 data available (over 5 140), the average requirement was 39.6 ± 3.7 MJ ME/d.
Other data have not been stored in an efficient way over the time to answer to this question surely and
rapidly within the 10 days allowed.
In the method section, can you please describe the culling criteria used in this farm? Because it is
relevant to longevity.
Answer: Sometimes too many sows are inseminated, and decision to cull the sow is not due to
lameness or health or any problem. In that situation, the data obtained on these sows were not
considered. L208-210
Data from sows that were supernumerary with regard to the number of available places in the
farrowing unit were removed from the analysis.
L166-168 Why was the population considered to be different every 4 years?
Answer: The aim was to check if the characteristics of the sows would change rapidly or not and if the
equation has to be updated. If required, every 4 years was a compromise to account for the time
required to collect data. If required, a shorter time interval may have been considered. As the answer
was no, it was kept at 4 years.
Table 1 number of sows in each backfat category is very small. Why was such a small number of sows
were selected for the analysis in Table 1?
Answer: the number is small because data were obtained only from 5 batches. No more batches were
managed under such hazardous conditions (as far as nutrition was considered) during the time
required to renew the facilities.
Table 1 The sows in <14 mm group gained bodyweight but reduced backfat (11.1 mm to 10.1 mm)
from d7 gestation to farrowing. Was it due to the small sample size used? Please double-check the
data.
Answer: They lost backfat (11.1 to 10.1 mm) AND bodyweight (232 to 223 kg) during lactation. And
they gain body weight (216 to 232 kg) BUT they lost backfat (13.7 to 11.1 mm) from day 7 to
farrowing. It was interpreted as the result of the competition in the access to the feed. The BW gain
was limited compared to other groups. It may correspond to the protein deposition (the driving force
mentioned earlier) or to another reason such as straw intake that was possible during this period.
L319 Should a farm update the equations to predict BWfinal more frequently if it does rigorous genetic
selection on backfat, litter size, and average birth weight in the breeding program?
Answer: Not necessarily, if the genetic selection focused strongly on litter size, as it increased
noticeably in the demonstration farm without a strong impact on BW curve. The answer will probably
differ about selection on body fatness, as it is known that selection for reduced fatness is associated
with a decreased maturity at a given age (that contributes to the necessity to revise the equation but
not every 4 years in our case study). What about the impact of genetic selection on average birth
weight? Maybe yes…

Reviewer 2 Report

The paper provides a series of interesting data relating to the long-term application of a precision feeding strategy based on an INRAE modeling approach.

Referring also to lactation performance, the Author must provide more details on lactating feeding and the management of lactating sows (and piglets).

Author Response

Rev2. comments:

The paper provides a series of interesting data relating to the long-term application of a precision feeding strategy based on an INRAE modeling approach.

Referring also to lactation performance, the Author must provide more details on lactating feeding and the management of lactating sows (and piglets).
The study did not focus on lactation performance, except when 5 groups of sows that differed by the BTfinal were compared.
L201-202 sentence added about them with sows fed ad libitum from the 5th day of lactation to the day before weaning.

Round 2

Reviewer 1 Report

Thank you for answering the questions.